# 2,2-difluorovinyl benzoates for diverse synthesis of *gem*-difluoroenol ethers by Ni-catalyzed cross-coupling reactions

Bingnan Du[1], Chun-Ming Chan [1,3], Pui-Yiu Lee [1,3], Leong-Hung Cheung [1], Xin Xu[2], Zhenyang Lin [2✉] & Wing-Yiu Yu [1✉]

*gem*-Difluoroalkene is a bioisostere of carbonyl group for improving bioavailability of drug candidates. Herein we develop structurally diverse 2,2-difluorovinyl benzoates (BzO-DFs) as versatile building blocks for modular synthesis of *gem*-difluoroenol ethers (44 examples) and *gem*-difluoroalkenes (2 examples) by Ni-catalyzed cross coupling reactions. Diverse BzO-DFs derivatives bearing sensitive functional groups (e.g., $C = C$, TMS, strained carbocycles) are readily prepared from their bromodifluoroacetates and bromodifluoroketones precursors using metallic zinc as reductant. With Ni(COD)$_2$ and dppf [1,1'-bis(diphenylphosphino)fer-rocene] as catalyst, reactions of BzO-DFs with arylboronic acids and arylmagnesium/alkyl-zinc reagents afforded the desired *gem*-difluoroenol ethers *and gem*-difluoroalkenes in good yields. The Ni-catalyzed coupling reactions features highly regioselective C(vinyl)–O (benzoate) bond activation of the BzO-DFs. Results from control experiments and DFT cal-culations are consistent with a mechanism involving initial oxidative addition of the BzO-DFs by the Ni(0) complex. By virtue of diversity of the BzO-DFs and excellent functional group tolerance, this method is amenable to late-stage functionalization of multifunctionalized bioactive molecules.

[1] State Key Laboratory of Chemical Biology and Drug Discovery and Department of Applied Biology and Chemical Technology, The Hong Kong Polytechnic University, Hung Hom, Kowloon, Hong Kong. [2] Department of Chemistry, The Hong Kong University of Science and Technology, Clear Water Bay, Kowloon, Hong Kong. [3] These authors contributed equally: Chun-Ming Chan, Pui-Yiu Lee. ✉email: chzlin@ust.hk; wing-yiu.yu@polyu.edu.hk

*g*em-Difluoroalkene functionality is considered as the bioisostere of a carbonyl group in drug design (Fig. 1a)[1,2]. Drug molecules containing carbonyl groups are prone to reduction by NAD(P)H-dependent reductases, causing poor bioavailability owing to drug-inactivation and excretion[3,4]. Bonnet–Delpon and co-workers already demonstrated that *gem*-difluoroalkene-derivatized artemisinins exhibit prolonged in vivo antimalarial activity through replacement of the carbonyl group[5]. Moreover, *gem*-difluoroalkenes can be transformed to various organofluorine motifs[6,7] such as trifluoromethyl[8–10], difluoromethyl[11–14], monofluoroalkenyl[15–18], and difluoromethylenyl[19,20] groups for late-stage functionalization. Conventionally, Wittig- or Horner-Wadsworth-Emmons-type olefination and $S_N2'$ reactions of vinyl trifluoromethyl compounds are widely employed to construct the *gem*-difluoroalkene moiety[7,21,22]. However, development of new strategies for structurally diverse synthesis of the *gem*-difluoroenol ethers and *gem*-difluoroalkenes under mild conditions is highly desirable.

Transition metal-catalyzed defluorinative cross-coupling reactions are attracting considerable attention for the synthesis of structurally diverse *gem*-difluoroalkenes (Fig. 1b)[23–26]. For instance, Hu[27] pioneered the Cu-catalyzed defluorinative coupling of trifluoromethylsilane and diazoalkanes. Alternatively, transition metal-mediated aryl/alkylative allylic β-fluoride elimination of trifluoromethyl alkenes also demonstrates enormous promises in this pursuit. In 2008, Murakami and co-workers reported the first example of *gem*-difluoroalkenes synthesis by the Rh-catalyzed addition of arylboronic acids to α-(trifluoromethyl) styrene, followed by β-fluoride elimination[28]. Later, the groups of Hayashi[29] and Lautens[30] independently developed the

analogous Rh-catalyzed asymmetric arylative β-fluoride elimination reactions. Recently, Ni-catalyzed defluorinative reductive cross-coupling of trifluoromethyl alkenes with N-hydroxyphthalimide esters[31], acetals[32], cyclobutanone oxime esters[33], and alkenes[34] have been reported. Norton and co-workers demonstrated the hydrodefluorination reactions of CF₃-substituted alkenes catalyzed by Ni(II)-hydride complexes. The reaction was mediated by H-atom transfer from Ni(II)-hydride, followed by defluorination[35].

While the current cross-coupling protocols are effective for the synthesis of *gem*-difluoroalkenes, other moieties such as *gem*-difluoroenol ethers which are also found in some pharmaceutical compounds remains less accessible[36–40]. In this regard, Katz and co-workers reported the preparation of stable potassium trifluoroborate *gem*-difluoroenol ethers, which would deliver the *gem*-difluoroenol ethers by cross-coupling with aryl halides under palladium catalysis[41]. To our knowledge, methods producing *gem*-difluorovinyl building blocks with immense structural diversity for the *gem*-difluoroenol ethers *and gem*-difluoroalkenes synthesis are rare. Herein we report a convenient synthesis of structurally diverse 2,2-difluorovinyl benzoates (BzO-DFs) by zinc treatment of the abundant bromodifluoroacetates and bromodifluoroketones as starting materials. The BzO-DFs serve as versatile building blocks for synthesis of diverse *gem*-difluoroenol ethers *and gem*-difluoroalkenes by Ni-catalyzed arylation and alkylation using nucleophiles such as arylboronic acids, arylmagnesium bromides, alkylzinc bromides, potassium alkyltrifluoroborates. Encouragingly, cross electrophile coupling reactions involving alkyl chlorides were also feasible under Ni catalysis (Fig. 1c).

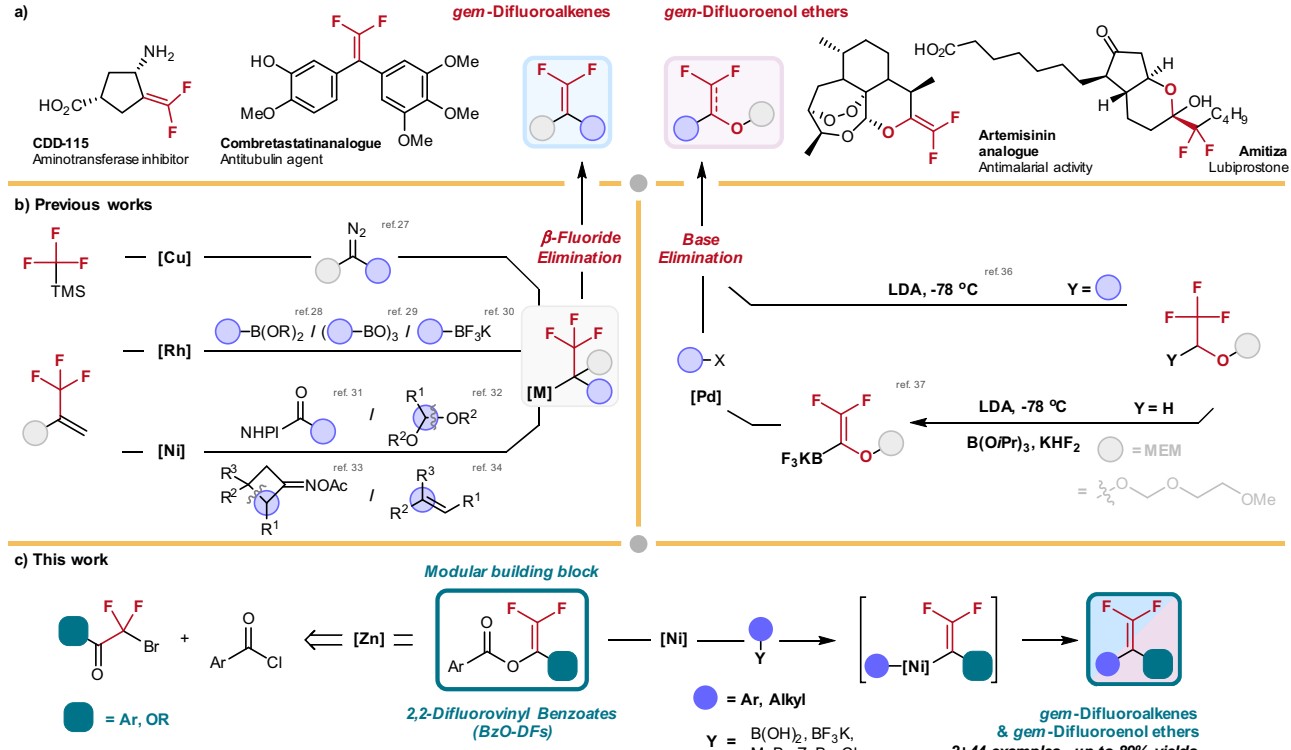

**Fig. 1 Cross-coupling approaches toward *gem*-difluoroenol ethers and *gem*-difluoroalkenes synthesis. a** Selected example of bioactive *gem*-difluoroenol ethers and *gem*-difluoroalkenes. **b** Conventional synthetic approaches for *gem*-difluoroenol ethers and *gem*-difluoroalkenes synthesis rely on β-fluoride elimination and base elimination as key steps; this approach demands tedious preparation of the trifluoromethyl-substituted precursors. **c** This work presents a convenient synthesis of 2,2-difluorovinyl benzoates (BzO-DFs) by simple zinc reduction of abundant bromodifluoroacetates and bromodifluoroketones; the BzO-DFs serve as versatile building blocks for Ni-catalyzed cross-coupling reaction for modular synthesis of *gem*-difluoroenol ethers and *gem*-difluoroalkenes.

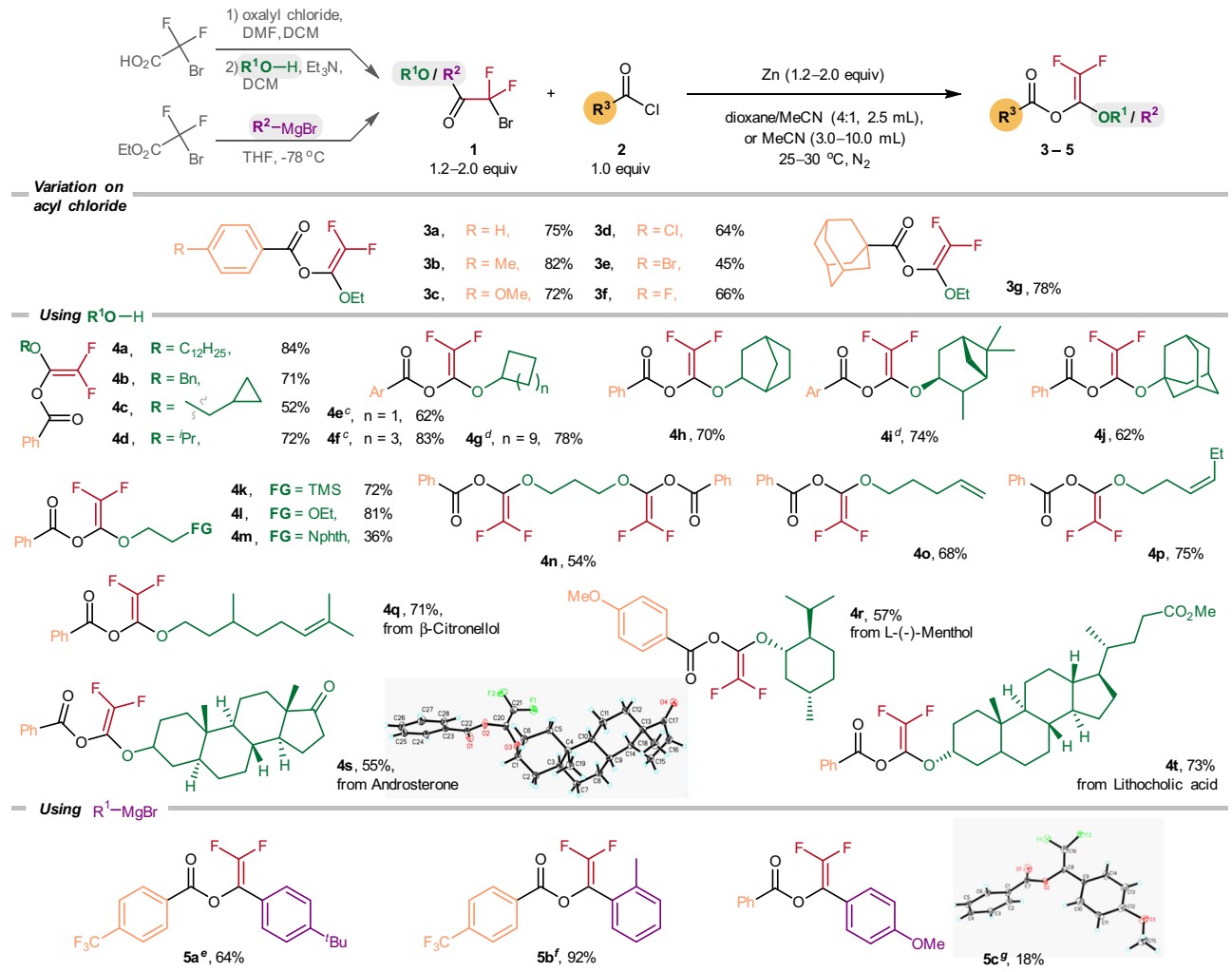

**Fig. 2 Scope of 2,2-difluorovinyl benzoates (BzO-DFs) 2.** [a]Reaction conditions: 2-bromo-2,2-difluoroacetates (2.0 equiv), acyl chlorides (0.5 mmol), and Zn (2.0 equiv) in 2.5 mL dioxane/MeCN (4:1 *v/v*) at 30 °C, 12 h, N₂. [b]Isolated yields. [c]Ar = Ph. [d]Ar = 4-MeOC₆H₄. [e]**1o** (3.0 equiv), 4-(trifluoromethyl) benzoyl chloride (0.2 mmol), Zn (3.0 equiv) in 3.0 mL MeCN at 25 °C, 10 h, N₂. [f]**1p** (1.5 equiv), 4-(trifluoromethyl)benzoyl chloride (0.2 mmol), Zn (2.0 equiv) in 3.0 mL MeCN at 25 °C, 10 h, N₂. [g]**1n** (1.2 equiv), benzoyl chloride (2.0 mmol), and Zn (1.2 equiv) in 10.0 mL MeCN at 25 °C, 5 h, N₂.

## Results

**Synthesis and scope of BzO-DFs**. To begin, 2-bromo-2,2-difluoroacetate (1.0 mmol) was treated with benzoyl chloride (0.5 mmol) and zinc powder (2.0 equiv) in a dioxane/MeCN (4:1 *v/v*) mixture under a N₂ atmosphere, and **3a** was obtained in 77% yield (see Supplementary Information for optimization study). For structure characterization, the ¹⁹F NMR spectrum of **3a** featured two characteristic doublet signals at −116 and −117 ppm, which is consistent with the *gem*-difluoroenol ether structure. The molecular structures of the BzO-DF derivatives **4s** and **5c** have been unambiguously established by X-ray crystallography.

Employing the Zn-mediated protocol, we accomplished the synthesis of the BzO-DFs involving diverse 2-bromo-2,2-difluoroacetates and carboxylic acid chlorides (Fig. 2). Reacting 2-bromo-2,2-difluoroacetate with various substituted aroyl chlorides (4-Me, 4-OMe, 4-Cl, 4-Br, and 4-F) afforded the corresponding benzoates (**3a**–**3f**) in 45–82% yields. Coupling of the 1-adamantanecarbonyl chloride with 2-bromo-2,2-difluoroacetate gave **3g** in 78% yield. The 2-bromo-2,2-difluoroacetates derived from simple primary alcohols such as *n*-dodecyl alcohol, benzyl alcohol, and cyclopropyl methanol were transformed to their

difluorovinyl benzoates in 84% (**4a**), 71% (**4b**), and 52% (**4c**) yields, respectively.

Those 2-bromo-2,2-difluoroacetates derived from secondary and tertiary alcohols including isopropanol (**4d**: 72%), cyclobutanol (**4e**: 62%), cyclohexanol (**4f**: 83%), cyclododecanol (**4g**: 78%), 2-*exo*-norbornanol (**4h**: 70%), 3-pinanol (**4i**: 74%), and 1-adamantanol (**4j**: 62%) were also effectively converted to their corresponding BzO-DFs.

BzO-DFs containing a silyl (**4k**: 72%), ether (**4l**: 81%), phthalimide (**4m**: 36%), and trimethylene glycol group (**4n**: 54%) were also prepared, albeit in moderate yields. Similarly, those BzO-DFs analogs containing C═C bonds (**4o**: 68%; **4p**: 75%) were also obtained. Interestingly, BzO-DFs derived from some bioactive natural products such as β-citronellol (**4q**: 71%), androsterone (**4s**: 55%), L-(-)-menthol (**4r**: 57%) and lithocholic acid (**4t**: 73%) were also prepared successfully.

For the synthesis of the aryl-substituted BzO-DFs derivatives, we found that the best results were obtained by pairing electron-poor 4-(trifluoromethyl)benzoyl chloride with the bromodifluoroketones bearing electron-rich aryl substituents; the corresponding **5a** (4-tert-butylaryl) and **5b** (2-methylaryl) can be obtained in moderate to excellent yields (64%, 92%). On a contrary,

**Table 1 Optimization of reaction conditions.**

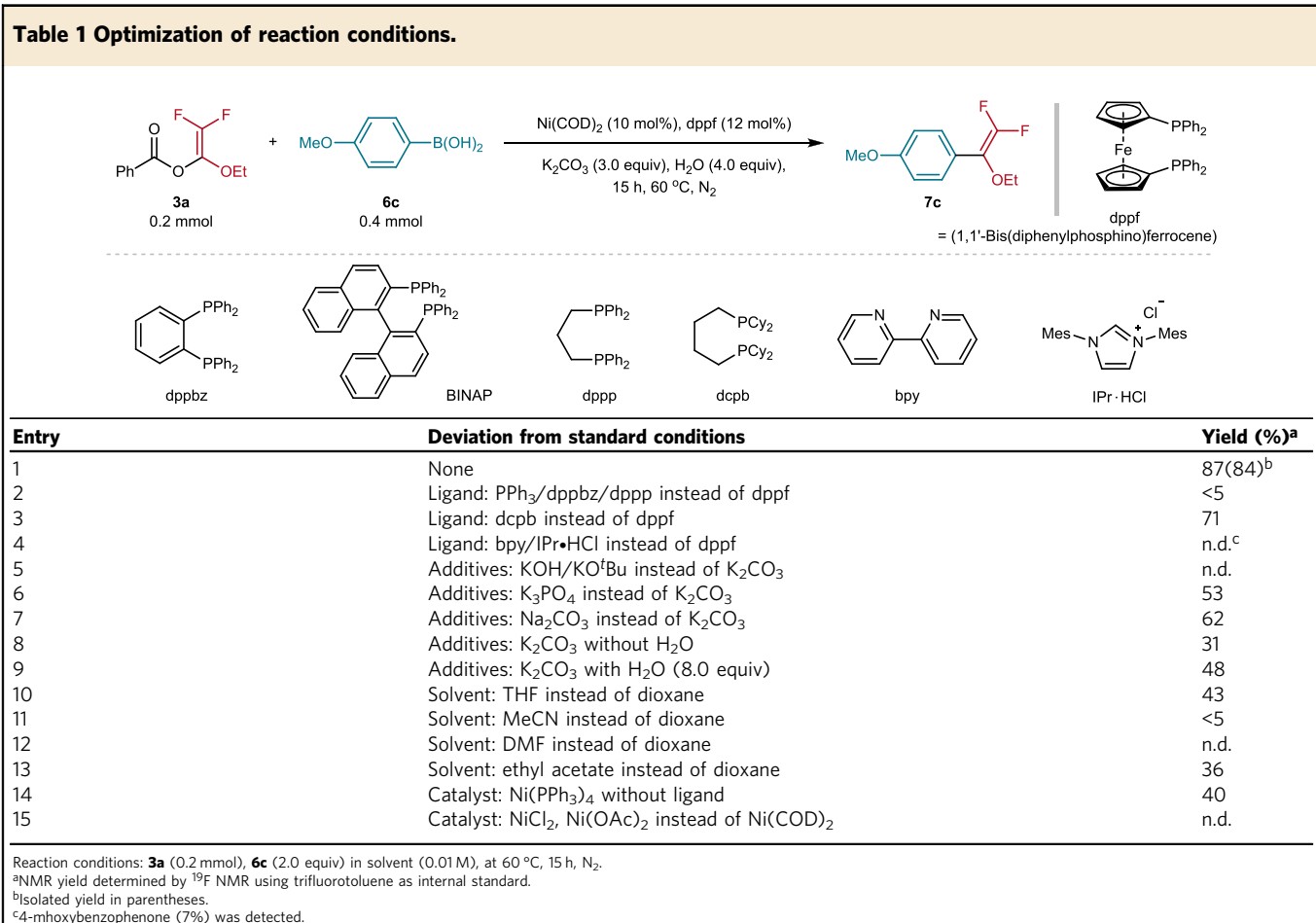

| Entry | Deviation from standard conditions | Yield (%)[a] |
|---|---|---|
| 1 | None | 87(84)[b] |
| 2 | Ligand: PPh₃/dppbz/dppp instead of dppf | <5 |
| 3 | Ligand: dcpb instead of dppf | 71 |
| 4 | Ligand: bpy/IPr•HCl instead of dppf | n.d.[c] |
| 5 | Additives: KOH/KO'Bu instead of K₂CO₃ | n.d. |
| 6 | Additives: K₃PO₄ instead of K₂CO₃ | 53 |
| 7 | Additives: Na₂CO₃ instead of K₂CO₃ | 62 |
| 8 | Additives: K₂CO₃ without H₂O | 31 |
| 9 | Additives: K₂CO₃ with H₂O (8.0 equiv) | 48 |
| 10 | Solvent: THF instead of dioxane | 43 |
| 11 | Solvent: MeCN instead of dioxane | <5 |
| 12 | Solvent: DMF instead of dioxane | n.d. |
| 13 | Solvent: ethyl acetate instead of dioxane | 36 |
| 14 | Catalyst: Ni(PPh₃)₄ without ligand | 40 |
| 15 | Catalyst: NiCl₂, Ni(OAc)₂ instead of Ni(COD)₂ | n.d. |

Reaction conditions: **3a** (0.2 mmol), **6c** (2.0 equiv) in solvent (0.01 M), at 60 °C, 15 h, N₂.
[a]NMR yield determined by $^{19}$F NMR using trifluorotoluene as internal standard.
[b]Isolated yield in parentheses.
[c]4-mhoxybenzophenone (7%) was detected.

the analogous reaction of benzoyl chloride and the 4-methoxyarylketone derivative afforded **5c** in only 18% yield.

**Reaction design and optimization.** While the Ni-catalyzed C–O bond activations[42,43] of aryl carbonates[44,45], aryl esters[46,47], and aryl carbamates[48,49] are well precedented, the analogous C–O bond activation of vinyl benzoates is less established[50]. Our next goal is to examine the Ni-catalyzed coupling reaction of BzO-DFs with arylboronic acids as nucleophiles through regioselective C (vinyl)–O(benzoate) bond activation. Pleasingly, when **3a** was treated with 4-methoxyphenylboronic acid **6c** in the presence of Ni(COD)₂ (COD = cyclooctadiene; 10 mol%), dppf (1,1'-bis (diphenylphosphino)ferrocene; 12 mol%), K₂CO₃ (3.0 equiv), and water (4.0 equiv) in dioxane (0.01 M) under an N₂ atmosphere, the desired *gem*-difluoroenol ether **7c** was obtained in 87% yield (Table 1, entry 1).

As shown in Table 1, the result with dppf as ligand is outstanding when compared to other common auxiliary ligands (Table 1, entry 2). Notably, the use of dcpb [1,4-bis(dicyclohexylphosphino)butane] furnished **7c** in 71% yield (Table 1, entry 3); yet bpy (2,2'-bipyridyl) and IPr·HCl (1,3-dimesityl-1H-imidazol-3-ium chloride) known to be utilized for some Ni-catalyzed cross couplings are completely ineffective (Table 1, entry 4).

The choice of bases was also critical; the use of strong bases such as KOH or KO'Bu led to decomposition of **3a** without **7c** being detected (Table 1, entry 5). Employing K₃PO₄ and Na₂CO₃ as bases gave **7c** in 53% and 62%, respectively (Table 1, entries 6–7). Apparently, 4.0 equiv of water as additive produced the best result. Reaction run under anhydrous conditions afforded **7c** in only 31% yield (Table 1, entry 8); see Supplementary Information

for details. The coupling was also sensitive to the reaction medium. When THF was used in lieu of dioxane, the yield of **7c** decreased significantly to 43% (Table 1, entry 10); the coupling reaction was largely suppressed when polar aprotic solvents such as MeCN, DMF and ethyl acetate were employed (Table 1, entries 11–13). Although Ni(PPh₃)₄ was catalytically active for this transformation with **7c** being formed in 40% yield (Table 1, entry 14); Ni(II) salts such as NiCl₂ and Ni(OAc)₂ were completely inactive with complete recovery of the starting materials (Table 1, entry 15).

**Substrate scope.** With the optimized conditions, we performed the cross-coupling scope study with a series of arylboronic acids (Fig. 3). For *para*-substituted arylboronic acids except those containing SMe (**7g**, 30%), OH (**7h**, 34%), and Cl (**7j**, 42%), the Ni-catalyzed coupling products were furnished in synthetically meaningful yields (**7a–7f**, **7i**). Arylboronic acids with coordinating nitrogen-, oxygen- and C=C moieties also reacted smoothly (**7k**: 52%; **7l**: 50%, **7m**: 40%). Similarly, successful couplings were also accomplished for the arylboronic acids with naphthalene motifs (**7n**: 76%; **7o**: 82%), disubstituted aryl groups (**7p**: 78%), and polyheterocyclic functions (**7q**: 74%; **7r**: 74%; **7s**: 56%; **7t**: 51%). Moreover, the coupling reactions have been applied for late-stage modification of some natural products compounds. For examples, the coupling reactions with the arylboronic acids derived from vitamin E, estrone and flavanone afforded the desired coupled products in moderate yields (**7u**: 26%; **7v**: 60%; **7w**: 39%).

Under the Ni-catalyzed conditions, BzO-DFs derived from simple primary alcohols (**8a–8c**: 66–89%) and several secondary

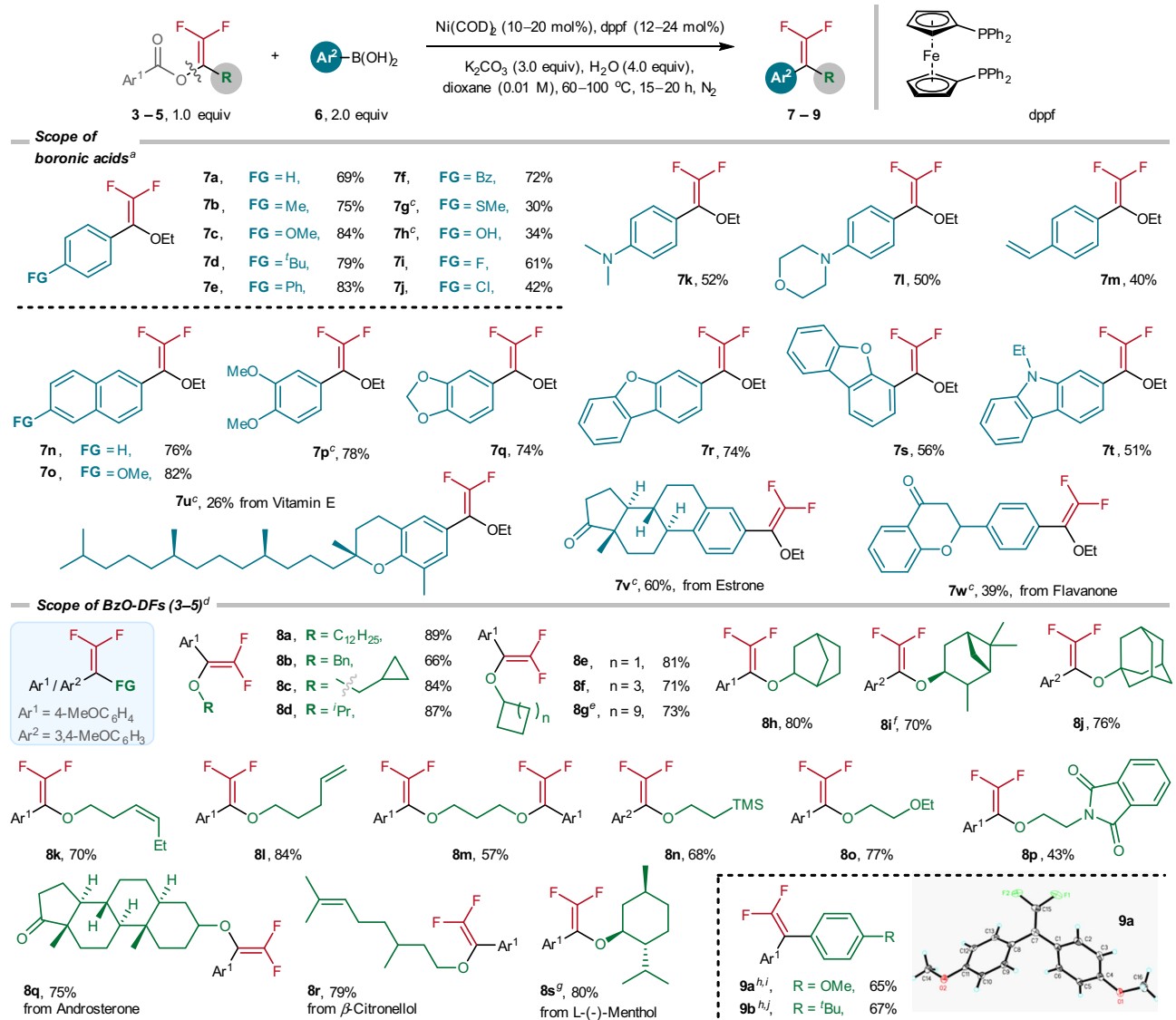

**Fig. 3 Ni-catalyzed cross-coupling reactions of BzO-DFs 3–5 and arylboronic acids 6.** [a]Reaction conditions: **3a** (0.2 mmol), **6** (2.0 equiv), Ni(COD)$_2$ (10 mol%), dppf (12 mol%), H$_2$O (4.0 equiv), K$_2$CO$_3$ (3.0 equiv) in dioxane (0.01 M) at 80 °C, 20 h, N$_2$. [b]Isolated yields. [c]Ni(COD)$_2$ (20 mol%), dppf (24 mol %), 80 °C, 20 h. [d]Reactions were performed at 60 °C for 15 h. [e]Using **4 g** instead of **3a**. [f]Using **4i** instead of **3a**. [g]Using **4r** instead of **3a**. [h]Ni(COD)$_2$ (20 mol %), dppf (24 mol%), 100 °C, 20 h. [i]Using **5c** instead of **3a**. [j]Using **5a** instead of **3a**.

and cycloalkanols (**8d**: 87%, **8e–8g**: 71–83%) were readily transformed to the corresponding coupled products. Gratifyingly, the benzoates with 2-*exo*-norbornanyl (**8h**: 80%), pinanyl (**8i**: 70%), and adamantyl (**8j**: 76%) were effectively arylated. Successful coupling reactions were achieved with benzoates derived from difunctionalized alcohols such as alkene (**8k**: 70%, **8l**: 84%), 1,3-propanediol (**8m**: 57%), trimethylsilyl ethanol (**8n**: 68%), ethereal (**8o**: 77%), and phthalimide moieties (**8p**: 43%) were well tolerated. The coupling reactions of 4-methoxyphenylboronic acid with the benzoates derived from androsterone (**8q**: 75%), β-citronellol (**8r**: 79%), and L-(-)-menthol (**8s**: 80%) were also successfully accomplished. In this work, two BzO-DFs derived from bromodifluoroketones (**1v**, **1x**) were also transformed to corresponding *gem*-difluoroalkenes (**9a**: 65%, **9b**: 67%). The molecular structure of **9a** has been established by X-ray crystallography.

**Synthetic applications**. We are gratified that the difluorovinyl benzoates synthesis and the subsequent Ni-catalyzed cross-

coupling reactions can be performed in a larger scale (2.0–6.0 mmol) without significant drop in yields (Fig. 4a). The practicability of BzO-DFs as the building blocks is also demonstrated by their favorable reactivity towards a wide range of coupling partners (Fig. 4b). Apart from arylboronic acids, aryl Grignard reagent was also found to be effective coupling partner. Employing Ni(COD)$_2$ (10 mol%) as catalyst and PPh$_3$ (20 mol%), **3a** reacted (4-methoxyphenyl)magnesium bromide to furnish **7c** in 45% yield (Fig. 4b, eq. 1). The current Ni-catalyzed reaction can be extended to C(sp$^2$)–C(sp$^3$) bond construction using some common alkyl-metal nucleophiles. For instance, the less-reactive *n*-hexylzinc bromide (Fig. 4b, eq. 2) and benzylzinc bromide (Fig. 4b, eq. 3) [dppf (24 mol%) as ligand)] with **4b/3a** gave the corresponding difluoroalkenes **11** and **10** in 38% and 22% yields, respectively.

In this work, we also examined the possibility to engage **3a** with some electrophiles for cross electrophile coupling reactions. We are pleased that benzyl chloride successfully coupled **3a** to give **10** in 47% yield using some conventional Ni-catalyzed reductive coupling conditions: Ni(OAc)$_2$ (10 mol%), bpy (12 mol%) and

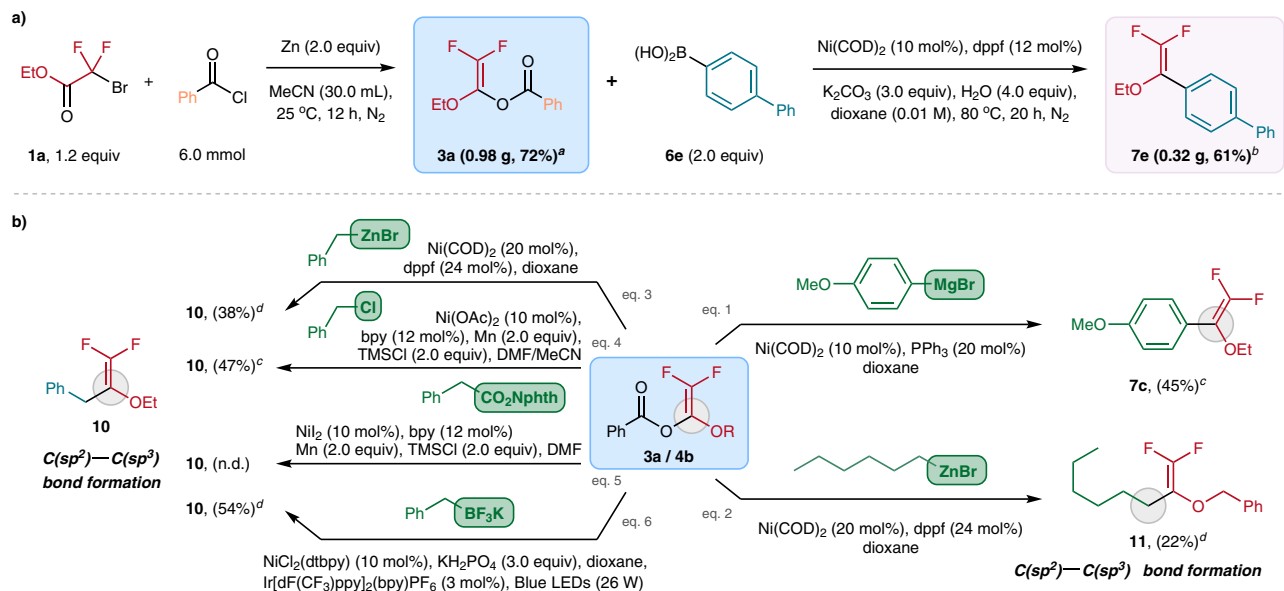

**Fig. 4 Synthetic applications. a** Gram-scale operation of the BzO-DFs synthesis and the subsequent Ni-catalyzed arylation. **b** Exploration of the coupling partners with the BzO-DF building blocks: catalytic protocols based on the uses of organometallic nucleophiles such as organomagnesium/-zinc and organotrifluoroborates as well as electrophiles such as benzyl chlorides have been developed. [a]**1a** (1.46 g) was used at starting material. [b]**3a** (0.46 g) was used for the Ni-catalyzed coupling reaction. [c]NMR yield determined by [19]F NMR with trifluorotoluene as internal standard. [d]Isolated yields.

Mn (2.0 equiv) (Fig. 4b, eq. 4). Redox-active ester failed to alkylate with **3a** under similar reductive coupling conditions (Fig. 4c, eq. 5). Apparently, our "Ni(COD)$_2$ + dppf" protocol is not effective when potassium benzyltrifluoroborate was employed as the coupling partner. Yet, effective C(sp$^2$)–C(sp$^3$) coupling was achieved under photochemical conditions. When we employed Molander's protocol [i.e., NiCl$_2$(dtbpy) (10 mol%), Ir[dF(CF$_3$) ppy]$_2$(bpy)PF$_6$ (3 mol%), blue light (26 W)], benzyltrifluoroborate reacted with **3a** to forge **10** formation in 54% yield (Fig. 4b, eq.6).

**Mechanistic studies.** Underlying the favorable reactivity of the BzO-DFs for *gem*-difluoroenol ethers synthesis is the regioselective C(vinyl)–O(benzoate) bond activation. Compared to the C (acyl)/C(alkyl)–O bonds activation, the observed selectivity may be attributed to the better leaving group reactivity of the benzoates[47]. In this regard, we performed a correlation study using a series of BzO-DFs (**3a/3b/3c/3d/3f/3h/3i**), which bear a series of substituted benzoic acids as leaving groups. Four sets of competition reaction: [**3a+3b**], [**3a+3c**], [**3a+3d**], [**3a+3f**], [**3a+3h**], [**3a+3i**] were performed, and the yields of remaining substrates were determined by [19]F NMR with PhCF$_3$ as an internal standard. Plotting log($k_{FG}/k_{H}$) with the Hammett σ(para) constants resulted a linear plot with a slope of +0.82 ($R^2 = 0.82$) of log($k_{FG}/k_{H}$) against the σ(para) constants. This finding indicated a faster reaction rate is promoted by electron-withdrawing substituents on the benzoates. This implies that partial negative charge build-up at the transition state of the oxidative addition.

In terms of product formation, it was found that the Ni-catalyzed arylation of **3a**–**3f** exhibited rather optimal yields (>80%) when the p$K_a$ values are within the range of 4.14–4.57. Lower yields were observed for **3d** (LG = 4-chlorobenzoic acid) and **3e** (LG = 4-bromobenzoic acid) despite of the lower p$K_a$ values of 3.97. (Fig. 5b). As expected, for the BzO-DF with adamantanecarboxylic acid (p$K_a$ = 4.86) as the leaving group, no coupling product was obtained under the standard conditions (see Supplementary Information for details). The benzoates were indeed the leaving group for the C(vinyl)–O activation; this is verified by the isolation of the benzoic acids in 76% yield after treating the reaction crude with HCl (aq) (Fig. 5c, eq. 1).

Regarding to the mechanism, Ni(0) complexes are known to effect C(sp$^2$)-halide bond activation initiated by single-electron transfer mechanism. It appeared plausible that the C(vinyl)–O (benzoate) activation may proceed via initial single-electron reduction of the BzO-DFs by the Ni(0) complex, leading to carboradicals formation via the C–O bond fragmentation. In this work, the radical mechanism was scrutinized by using TEMPO as radical trap. When the "**3a**+**6c**" reaction was performed in the presence of TEMPO (2.0 equiv), the **7c** formation was completed suppressed with 61% recovery of the starting **3a** (Fig. 5c, eq. 2). Apparently, no TEMPO-radical adducts were detected by GC-MS and LC-HRMS analysis of the reaction mixture. The lack of product formation could be attributed to the oxidation of the active Ni(0) complexes by the TEMPO, resulting in the loss of the catalytic activity. Consistent with this notion, when BHT (2.0 equiv) was used in place of TEMPO as radical trap, effective formation of **7c** (85%) was detected by NMR analysis (Fig. 5b, eq. 3). Furthermore, when **3a** and **6c** were subjected to the Ni-catalyzed conditions in the presence of α-cyclopropylstyrene (2.0 equiv) as radical probe, **7c** was formed in 75% yield and the radical probe was completely recovered unchanged (Fig. 5c, eq. 4). On the basis of these findings, pathways involving radical formation is untenable.

We performed density-functional theory (DFT) calculation to delineate the mechanistic details of the reaction. We chose the "**3a** +**6c**" reaction as a model for our DFT study. Based on our ligand screening study, we believed that the dppf-ligated Ni(0) species **A** is the active species to initiate the coupling reaction. Oxidative addition of **3a** should pass through a barrier of ~27.2 kcal mol$^{-1}$. This step is found to be the rate-determining step[51,52]. As shown in Fig. 6, oxidative addition of **3a** to form complex **B** is nearly a thermal neutral process. Yet, the major driving force of the oxidative addition appears to be the ligand exchange reaction of the benzoate with K$_2$CO$_3$ to form complex **C**, which is highly exergonic with a drop of the relative Gibbs free energy from 2.0 kcal mol$^{-1}$ to −12.4 kcal mol$^{-1}$. Subsequent transmetalation with arylboronic acid **6c** takes place via a six-membered ring transition state (**TS$_{C-D}$**) to generate intermediate **D**[53,54]. The transmetalation is followed by spontaneous reductive elimination to give Ni

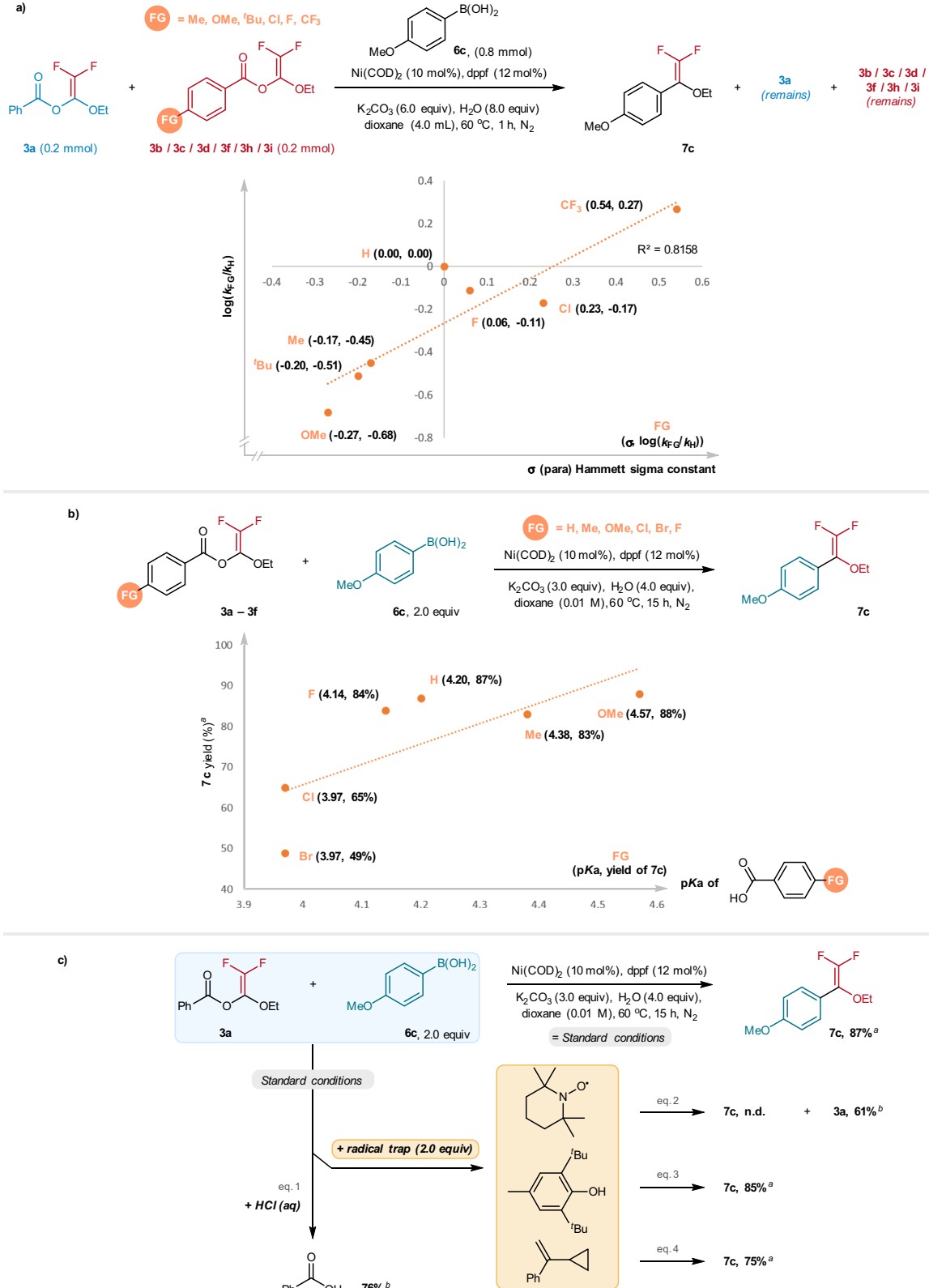

**Fig. 5 Mechanistic studies. a** Hammett correlation study of the Ni-catalyzed cross-coupling reactions of BzO-DFs **3** and arylboronic acids 6**a**. **b** Effects of leaving group on the **7c** formation. **c** Probing the radical mechanism. [a]NMR yield determined by [19]F NMR with trifluorotoluene as internal standard. [b]Isolated yields.

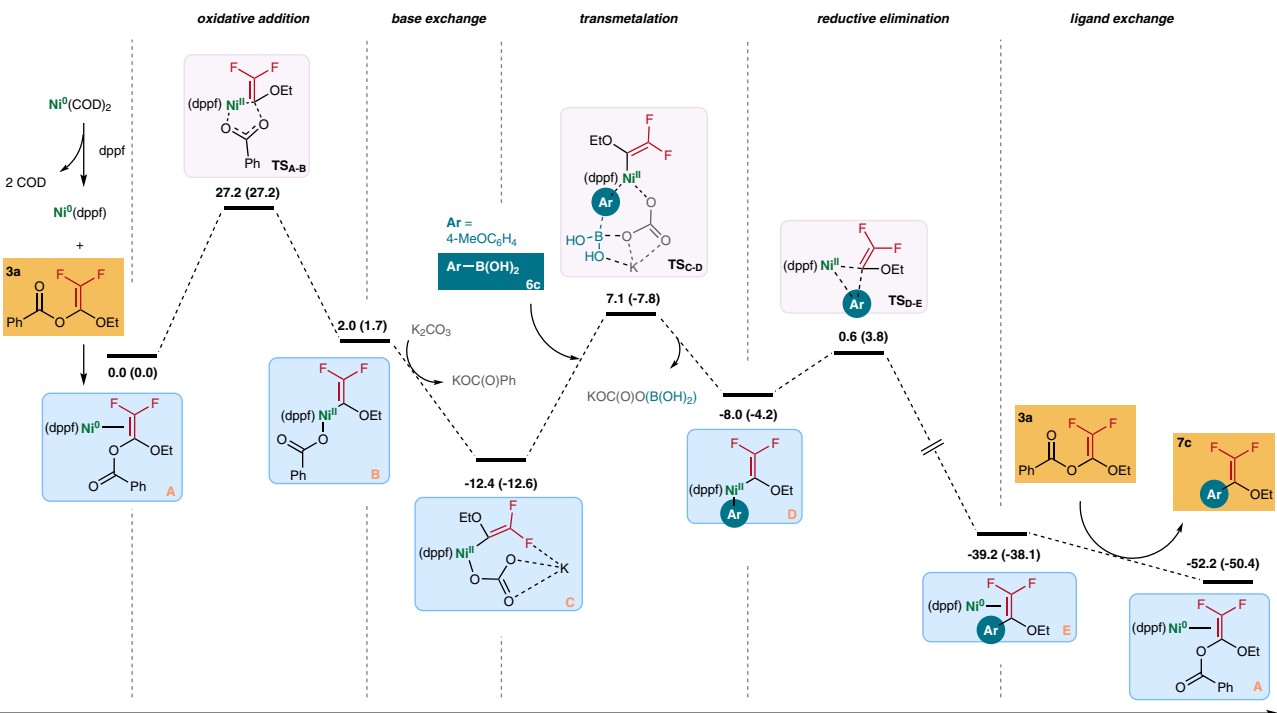

**Fig. 6 Gibbs free energy profile calculated for the Ni-catalyzed reaction of 3a and 6c.** The relative Gibbs energies and Electronic energies (in parentheses) are given in kcal/mol.

(0) complex **E** involving a drop of ΔG of the system from −8.0 kcal mol$^{-1}$ to −39.2 kcal mol$^{-1}$. Finally, **E** reacts with **3a** through ligand exchange to give product **7c** with the regeneration of the active species **A**, which is very facile with low energy barriers.

In summary, we developed a modular synthesis of *gem*-difluoroenol ethers *and gem*-difluoroalkenes by the Ni-catalyzed coupling with structurally diverse BzO-DFs. The BzO-DFs can be easily prepared from abundant bromodifluoroketones and bromodifluoroacetates. Exhibiting remarkable functional groups tolerance, this Ni-catalyzed protocol permits easy installation of a *gem*-difuorovinyl group to bioactive molecules. This reaction should supplement the conventional strategies in exploiting difluorovinyl moieties in drug design and development. Control experiments and DFT calculations revealed that the reaction is initiated by regioselective oxidative addition of the BzO-DFs by the Ni(0) complex. The intermediate vinylnickel(II) benzoate complexes readily exchanged with organometallic nucleophiles enabling a versatile C(sp$^2$)–C(sp$^2$) and C(sp$^2$)–C(sp$^3$) coupling manifolds.

## Methods

**General procedure for preparation of BzO-DFs (3–5)**. To a 25 mL round-bottom flask with a magnetic stir bar was added acyl chloride (1.0 equiv, 0.5 mmol), Zn powder (2.0 equiv, 1.0 mmol), MeCN (0.5 mL) and dioxane (1.5 mL) in a nitrogen-filled glovebox. The reaction mixture was then sealed with a rubber septum and removed from the glovebox. To the mixture, **1** (2.0 equiv, 1.0 mmol) was diluted with dioxane (0.5 mL) and added with a syringe pump over 20 min. After stirring the reaction mixture for further 12 h at 30 °C, the reaction mixture was filtered, and the filtrate was concentrated in vacuo. The residue was purified by flash column chromatography (*n*-hexane:ethyl acetate = 200:1–50:1) to give the desired 2,2-difluorovinyl benzoates.

**General Procedure for Ni-catalyzed cross-coupling reaction between BzO-DFs (3–5) and arylboronic acids**. To a 8 mL vial equipped with a magnetic stir bar was added 2,2-difluorovinyl benzoate **3–5** (1.0 equiv, 0.2 mmol), arylboronic acid **6** (2.0 equiv, 0.4 mmol), K$_2$CO$_3$ (3.0 equiv, 0.6 mmol) and deionized water (4.0 equiv, 0.8 mmol) in a nitrogen-filled glovebox. Then a premixed solution of Ni(COD)$_2$ (10 mol %, 0.02 mmol), dppf (12 mol%, 0.024 mmol) and dioxane (0.01 M, 2.0 mL) was transferred to the reaction mixture. The 8 mL vial was then capped and removed from the glovebox. After stirring the reaction mixture for 15 h at 60 °C, the reaction mixture

was cooled to room temperature and diluted with ethyl acetate. The solution was then filtered through a pad of Celite®. The filtrate was then dried over MgSO$_4$ and concentrated in vacuo. The residue was purified by flash column chromatography (*n*-hexane:ethyl acetate = 200:1–50:1) to give the desired products.

## Data availability

All data are available from the corresponding authors upon reasonable request. The X-ray crystallographic coordinates for structures reported in this study have been deposited at the Cambridge Crystallographic Data Centre (CCDC), under deposition numbers 2008864 (**4s**), 1921828 (**5c**), and 1921829 (**9a**). These data can be obtained free of charge from The Cambridge Crystallographic Data Cenrte via www.ccdc.cam.ac.uk/data_request/cif".

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

## Acknowledgements

We acknowledge the financial support of The Hong Kong Research Grants Council (153037/14P, 153152/16P, 153023/17P, 153017/19P, and C5023-14G). Dr. Bingnan Du and Dr. Chun-Ming Chan thanks The State Key Laboratory for Chemical Biology and Drug Discovery for the generous support of the Postdoctoral Fellowship.

## Author contributions

B.D. performed the experiments. W.-Y.Y. conceived, directed the project, and wrote the paper. C.-M.C. engaged in revision of the manuscript, mechanistic studies, revamping the data, and graphical presentation of the manuscript and revised the Supplementary Information. P.-Y.L. and L.-H.C. prepared and organized physical characterization data, involved in discussion of mechanism, purification of substrates and preparation of the initial draft of the manuscript. X.X. and Z.L. conducted the density functional theory calculations.

## Competing interests

The authors declare no competing interests.
