## [Peer Review File · Nature Communications]

Reviewers' comments:

Reviewer #1 (Remarks to the Author):

In the submitted manuscript, Yu and co-workers described a modular strategy for the synthesis of gem-difluoroalkenes through Zn mediated preparation of 2,2-difluorovinyl benzoates (BzO-DFs) and subsequent nickel-catalyzed C-C coupling of the BzO-DFs with nucleophiles. The preparation of BzO-DFs was convenient from bromodifluoroacetates under simple reductive conditions. The nickel-catalyzed C-C coupling was carried out under mild conditions and tolerated a wide range of synthetic important functional groups. Considering the importance and applications of gem-difluorovinyl ethers in agrochemistry and medicinal chemistry and also the existing methods, this new strategy has some advantages and deserves more attentions. I personally support the publication of this manuscript on Nature Communications.

Following are my suggestions for minor revisions:

1) It will be delighted to see the X-ray crystallographic structure of compound 2ud. It was a white solid.

2) The yields for substrates containing Cl (4ae, 42%), SMe (4ai, 30%), and OH (4aj, 34%) were much lower. Which side reactions were observed for boronic acids and BzO-DFs?

3) There are two nice examples of benzyl nucleophiles in Figure 5. How about the reactivity of unactivated alkyl nucleophiles, such as n-Hexyl-ZnBr or n-Hexyl-BF₃K, in the step of nickel-catalyzed C-C coupling?

4) Important literatures for the synthesis of gem-difluoroalkenes should be added. a) Molander realized the defluorinative alkylation of trifluoromethyl alkenes using radical precursors under photocatalysis conditions, *Angew. Chem. Int. Ed.* 2017, 56, 15073. b) Zhou reported the photocatalytic decarboxylative/defluorinative reaction for the synthesis of functionalized gem-difluoroalkenes, *J. Org. Chem.* 2016, 81, 7908. c) Recent review for selective single C-F activation in trifluoromethyl groups, *Chem. Eur. J.* 2018, 24, 14572.

5) In the work of Lu and Fu (ref. 27, *Chem. Sci.* 2019, 10, 809), not only redox-active esters were suitable alkylation reagents, but also alkyl bromides. Carboxylic acid derivatives have a different source compared with alkyl halides. Both of them are important alkyl electrophiles.

6) I don't agree with the words 'radical defluorinative coupling' in this manuscript. I think Hashmi's work (*Angew. Chem. Int. Ed.* 2016, 55, 9416) might be described as 'radical defluorinative coupling'. Lu and Fu's work should be called 'defluorinative reductive cross-coupling', and this concept was put forward in 2017 for the first time by the same group (*J. Am. Chem. Soc.* 2017, 139, 12632). I suggest to add the descriptions of 'defluorinative reductive cross-coupling' and related references to the text.

Xi Lu

Reviewer #2 (Remarks to the Author):

This paper reports results that are interesting from the synthetic standpoint, but it is not well written and should not be published in its present form. We are willing to examine a rewritten version, but we are quite concerned about the issues in the next paragraph.

The radical experiments contradict each other, and the "relative reactivity" experiments do also. On p 6 line 149-151, "when the "2ad+3a" reaction ... the starting materials", the fact that TEMPO suppresses the reaction suggests a radical mechanism and is thus contradictory to the vinyl cyclopropane result. On p 5 line 134-138, "Moreover, the relative reactivity Slower than simple benzoate" does not make sense; the authors get contradictory results but do not offer an explanation, and line 138-140 "when the reaction crude ... leaving group for this reaction" does not make sense.

Minor Issues:

Page 2 line 39: "dialkyl-substituted" is not correct. Fig. 1b also shows the defluorinative arylation of CF₃ alkenes.

Page 3 line 62: "cyclopropyl alcohol" should be "cyclopropyl methanol"

Page 3 table 1: better explain the ligand abbreviations (dppf, but especially dppbz, dcpb)

Page 3 table 1, entries 14 & 16, it is not clear what "(0.5)" this means

Page 4 line 82: "is" should be "are"

Page 4 fig. 3: is R in substrate 2 always phenyl? If that's the case, it's better to just say phenyl

Page 4 fig. 3 line 97: "4.0 equiv of H₂O, 3.0 eq of K₂CO₃"; "eq" stands for equation, not equivalents

Page 4 fig. 3: for product 4sa, why is the corresponding substrate 2 not shown in Fig. 2?

Page 5 fig. 4: the results of Figure 4a should be incorporated into Figure 2

Page 6 line 154-157: the reaction with aryl Grignard reagent is not sp²-sp³ coupling.

Page 6 line 160: "oxidative addition with 3" should be "oxidative addition with 2"

Page 6 Fig. 5: the very last reaction: why is a photocatalyst needed?

Page 7 line 183: "a magnetic stir" should be "a magnetic stir bar"

A paper that uses a different Ni(II) catalyst to prepare gem difluoroalkenes (from a different substrate) has recently appeared: pubs.acs.org/doi/10.1021/jacs.9b13757

Reviewer #3 (Remarks to the Author):

The publication "Modular Approach for Synthesis of gem-difluoroalkenes by Ni-Catalyzed Cross Coupling of Various Nucleophiles with 2,2-Difluorovinyl Benzoates" by Yu et al. describes a metal-catalyzed route to difluoro alkenes. The concept is interesting and the authors have done an excellent job of illustrating the wide range of ether substituents that can be employed. The subsequent use of the benzoates in cross coupling approaches is an excellent use of the initially prepared benzoates.

Although the authors note that their work describes a synthesis of difluoro alkenes, this is not really true. In fact it is only applicable to difluoro enol ethers, which is a very specific class of difluoroalkene. In the introduction, the authors describe several other examples of the use of cross coupling to prepare difluoro alkenes and they also note that Wittig-type reactions can be used to prepare these compounds. The authors then note that difluoro enol ethers are "largely inaccessible" by these routes. However there are many other ways to prepare difluoro enol ethers of exactly the type described herein. There are no references to other methods to prepare this functional group. Examples such as the use of CF₂Br₂ with lactones, or the reaction of BrCF₂CO₂R with Mg or Zn, which is related to the current work. Base mediated elimination from CF₃CH(OR)R type systems was not discussed. In addition, the authors did not comment on how their method would compare with simply making a difluoro enol triflate from the corresponding ketone and having it participate in a cross coupling reaction.

Thus although the work was clearly well carried out, the novelty of the method has not been demonstrated to a level sufficient for publication in a journal like Nature Communications. In terms of detailed scientific advice, I was surprised that the authors did not perform a full Hammett plot to aid in their comparison of electronic effects. I also noted that although the authors claimed that the reaction of compound 2 was not affected by the substituents, both Cl and Br-containing

compounds reacted with considerably lower yields, which may suggest competing cross coupling occurring at the aryl halides. This should be investigated

Therefore it is the opinion of this reviewer that although the work is technically sound, it is not of sufficient novelty or synthetic utility relative to other methods for publication in Nature Communications.

Point by point response to the reviewer's comments (NCOMMS-20-07556)

Reviewer: 1

It will be delighted to see the X-ray crystallographic structure of compound 2ud. It was a white solid.

The compound previously labelled as **2ud** has its molecular structure characterized by X-ray crystallography (see figure below). Please note that the numbering system has been revamped, and the compound **2ud** is now labelled as **4s**. The crystallographic structure is added to Fig 2.

The yields for substrates containing Cl (4ae, 42%), SMe (4ai, 30%), and OH (4aj, 34%) were much lower. Which side reactions were observed for boronic acids and BzO-DFs?

With our revised labeling system, please note the transformation: (**4ae** → **7j**), (**4ai** → **7g**), and (**4aj** → **7h**).

To address this concern, we performed side-products analysis for the Ni-catalyzed cross-coupling reaction of BzO-DF (**3a**) with (4-chlorophenyl) / (4-(methylthio)phenyl)- / (4-hydroxyphenyl)boronic acid. It was found that around 28–33% of **3a** remained under the optimized reaction conditions with deboronation (22–26%) of the arylboronic acids being observed.

There are two nice examples of benzyl nucleophiles in Figure 5. How about the reactivity of unactivated alkyl nucleophiles, such as *n*-Hexyl-ZnBr or *n*-Hexyl-BF₃K, in the step of nickel-catalyzed C–C coupling?

Thank you for the kind suggestion. An example using *n*-hexyl-ZnBr as alkyl nucleophile has been added (Fig. 4, eq. 2) in the revised manuscript and the corresponding alkylated product **11** was formed in 22% yield.

Important literatures for the synthesis of gem-difluoroalkenes should be added. a) Molander realized the defluorinative alkylation of trifluoromethyl alkenes using radical precursors under photocatalysis conditions, *Angew. Chem. Int. Ed.* 2017, 56, 15073. b) Zhou reported the photocatalytic decarboxylative/defluorinative reaction for the synthesis of functionalized gem-difluoroalkenes, *J. Org. Chem.* 2016, 81, 7908. c) Recent review for selective single C-F activation in trifluoromethyl groups, *Chem. Eur. J.* 2018, 24, 14572.

Thank you for the suggestions. The recommended citations have been added in this revised version (ref. 24–26).

In the work of Lu and Fu (ref. 27, Chem. Sci. 2019, 10, 809), not only redox-active esters were suitable alkylation reagents, but also alkyl bromides. Carboxylic acid derivatives have a different source compared with alkyl halides. Both of them are important alkyl electrophiles.

Thank you for the suggestion, two examples were added in the manuscript involving the use of benzyl chloride and redox-active ester as coupling partners. We are pleased to report that benzyl chloride successfully coupled with **3a** to give **10** in 47% yield under conventional Ni-catalyzed reductive coupling conditions: Ni(OAc)₂ (10 mol%), bpy (12 mol%), Mn (2.0 equiv), DMF (0.001 M), CH₃CN (0.004 M), TMSCl (2.0 equiv), 60 °C, 12 h, N₂. Yet, the redox-active ester failed to couple with **3a** under similar conditions. (Fig. 4, eq. 4–5).

I don't agree with the words 'radical defluorinative coupling' in this manuscript. I think Hashmi's work (Angew. Chem. Int. Ed. 2016, 55, 9416) might be described as 'radical defluorinative coupling'. Lu and Fu's work should be called 'defluorinative reductive cross-coupling', and this concept was put forward in 2017 for the first time by the same group (J. Am. Chem. Soc. 2017, 139, 12632). I suggest to add the descriptions of 'defluorinative reductive cross-coupling' and related references to the text.

We agree with this suggestion and the expression “radical defluorinative coupling” has been changed to “defluorinative reductive cross-coupling”. The suggested citation is also included (ref. 18).

Reviewer: 2

This paper reports results that are interesting from the synthetic standpoint, but it is not well written and should not be published in its present form. We are willing to examine a rewritten version, but we are quite concerned about the issues in the next paragraph.

The manuscript has been extensively rewritten with all the graphic presentations being redesigned for easier reading and comprehension. We thank the referee for this remark.

The radical experiments contradict each other, and the "relative reactivity" experiments do also. On p 6 line 149-151, "when the "2ad+3a" reaction ... the starting materials", the fact that TEMPO suppresses the reaction suggests a radical mechanism and is thus contradictory to the vinyl cyclopropane result.

Thank you for the advice, we performed a series of radical trapping experiments with some radical scavengers. The results were added to the manuscript: [When the "3a + 6c" reaction was performed in the presence of TEMPO (2.0 equiv), the 7c formation was completely suppressed with 61% recovery of the starting 3a (eq. 2). Apparently, no TEMPO-radical adducts were detected by GC-MS and LC-HRMS analysis of the reaction mixture. The lack of product formation could be attributed to the oxidation of the active Ni(0) complexes by the TEMPO, resulting in the loss of the catalytic activity. When BHT (2.0 equiv) was used in place of TEMPO as radical trap, effective formation of 7c (85%) was detected by NMR analysis (eq. 3). Furthermore, when 3a and 6c were subjected to the Ni-catalyzed conditions in the presence of α -cyclopropylstyrene (2.0 equiv) as radical probe, 7c was formed in 75% yield and the radical probe was completely recovered unchanged (eq. 4). On the basis of these findings, pathways involving radical formation is untenable, which is consistent with our DFT calculation.]

On p 5 line 134-138, “Moreover, the relative reactivity Slower than simple benzoate” does not make sense; the authors get contradictory results but do not offer an explanation, and line 138-140 “when the reaction crude ... leaving group for this reaction” does not make sense.

Given the data available at this juncture, the finding that simple benzoate exhibits higher relative reactivity than other substituted benzoates is difficult to be rationalized. Yet, this finding has been verified by repeating the experiments during our revising this manuscript. We opined that this peculiar phenomenon should warrant further investigation. While this investigation is ongoing, we decided to remove the relevant data from the manuscript and a more detailed mechanistic study should be reported separately. We consider that removing this section would not compromise the significance and validity of the findings reported in this manuscript.

Minor Issues:

- **Page 2 line 39: “dialkyl-substituted” is not correct. Fig. 1b also shows the defluorinative arylation of CF_3 alkenes.**
 - Well noted and correction has been done.
- **Page 3 line 62: “cyclopropyl alcohol” should be “cyclopropyl methanol”**
 - Well noted and correction has been done.

- **Page 3 table 1: better explain the ligand abbreviations (dppf, but especially dppbz, dcpb)**

- Structures and full names of ligands used were added in the manuscript, (Table 1 and corresponding description)

- **Page 3 table 1, entries 14 & 16, it is not clear what "(0.5)" this means**
 - Table 1 has been revised.
- **Page 4 line 82: "is" should be "are"**
 - Well noted and correction has been done.
- **Page 4 fig. 3: is R in substrate 2 always phenyl? If that's the case, it's better to just say phenyl**
 - All the figures have been redesigned for better presentation and comprehension.
- **Page 4 fig. 3 line 97: "4.0 equiv of H₂O, 3.0 eq of K₂CO₃"; "eq" stands for equation, not equivalents**
 - Well noted and correction has been done.
- **Page 4 fig. 3: for product 4sa, why is the corresponding substrate 2 not shown in Fig. 2?**
 - All the figures have been redesigned, and the corresponding structures have been added for increased clarity.
- **Page 5 fig. 4: the results of Figure 4a should be incorporated into Figure 2**
 - Well noted and correction has been done.
- **Page 6 line 154-157: the reaction with aryl Grignard reagent is not sp²-sp³ coupling.**

- Well noted and the paragraph was revised.
- **Page 6 line 160: “oxidative addition with 3” should be “oxidative addition with 2”**
 - Well noted and correction has been done.
- **Page 6 Fig. 5: the very last reaction: why is a photocatalyst needed?**
 - We adopted the procedure reported by Molander and coworkers for photocatalytic radical coupling reactions. It is believed that the excited photocatalyst would oxidize the benzyl-BF₃K to form benzyl radicals, and the reduced photocatalyst would reduce [Ni^I] to [Ni⁰] for regenerating the active Ni catalyst for turnovers.
- **Page 7 line 183: “a magnetic stir” should be “a magnetic stir bar”**
 - Well noted and correction has been done.
- **A paper that uses a different Ni(II) catalyst to prepare gem difluoroalkenes (from a different substrate) has recently appeared: pubs.acs.org/doi/10.1021/jacs.9b13757.**
 - Well noted and recommended citation has been added to the reference section (ref. 35).

Reviewer: 3

Although the authors note that their work describes a synthesis of difluoro alkenes, this is not really true. In fact it is only applicable to difluoro enol ethers, which is a very specific class of difluoroalkene. In the introduction, the authors describe several other examples of the use of cross coupling to prepare difluoro alkenes and they also note that Wittig-type reactions can be used to prepare these compounds. The authors then note that difluoro enol ethers are "largely inaccessible" by these routes. However there are many other ways to prepare difluoroenol ethers of exactly the type described herein. This are no references to other methods to prepare this functional group. Examples such as the use of CF_2Br_2 with lactones, or the reaction of $BrCF_2CO_2R$ with Mg or Zn, which is related to the current work. Base mediated elimination from $CF_3CH(OR)R$ type systems was not discussed. In addition, the authors did not comment on how their method would compare with simply making a difluoroenol triflate from the corresponding ketone and having it participate in a cross coupling reaction.

Indeed, this work mainly focuses on the construction of difluoroenol ethers. Yet, this reaction can also be extended to preparation of difluoroalkenes (**5a**, **5b**, **5c** and **9a**, **9b**).

The reviewer is right to point out that there are several reports on the synthesis of difluoroenol ethers, and these works are also included in the reference section. Yet, those reported methods employ the classical transformation manifolds such as Wittig-type difluoromethylation of lactones with CF_2Br_2 (*ref. 13*) and base mediated elimination from $CF_3CH(OR)R$ type (*ref. 36*). These methods involve non-modular approach and are limited in product diversity. In this work, the BzO-DFs are designed as building blocks to enable modular synthesis of both difluoroenol ethers and difluoroalkenes by Ni-catalyzed cross coupling reaction. We are aware of a report by Katz and coworkers concerning the use of stable potassium trifluoroborate *gem*-difluoroenol ethers as building block for synthesis of *gem*-difluoroenol aryl ethers by Pd-catalyzed cross coupling with aryl halides. This work has been cited in *ref. 41* of the

revised manuscript.

Thus although the work was clearly well carried out, the novelty of the method has not been demonstrated to a level sufficient for publication in a journal like Nature Communications. In terms of detailed scientific advice, I was surprised that the authors did not perform a full Hammett plot to aid in their comparison of electronic effects. I also noted that although the authors claimed that the reaction of compound 2 was not affected by the substituents, both Cl and Br-containing compounds reacted with considerably lower yields, which may suggest competing cross coupling occurring at the aryl halides. This should be investigated

As noted earlier, the higher relative reactivity of simple benzoic acid than the substituted analogues is a subject of a separate investigation. We agreed with the reviewer that a full range Hammett study should be part of our on-going mechanistic study.

In this work, we reported the effects of pKa values of the leaving groups. It was found that optimal yields were observed for the pKa range of 4.14–4.57.

FG	pKa of FG-C ₆ H ₄ COOH	yield of 7c (%)
OMe	4.57	88
Me	4.38	83
H	4.20	87
F	4.14	84
Cl	3.97	65
Br	3.97	49

As noted by the reviewer, the synthesis of BzO-DFs bearing Cl- (**3d**), and Br- (**3e**) substituents came with lower product yields (45–64%). Further analysis of these reaction runs showed that side-products (**3d'**, **3d''**) and (**3e'**, **3e''**) have been formed.

REVIEWER COMMENTS

Reviewer #1 (Remarks to the Author):

I appreciate the authors' efforts to address the comments made by all reviewers in the previous round of review. The manuscript appears to be more appropriate for publication on Nature Communications at this point.

Xi Lu

Reviewer #2 (Remarks to the Author):

I think that the authors have done a really thorough revision of the original manuscript. Our comments are well addressed. I think that the present version should be published, after they fix the following minor typos.

Figure 1c: in the middle, "Y = B(OH₂)" should be "Y = B(OH)₂"

Page 9, Method: line 17: "stirrer bar" should be "stir bar"

Reviewer #3 (Remarks to the Author):

I have reviewed the revised document and remain unconvinced as to the Authors statement that this orwk corresponds to a diverse synthesis of gem difluoroalkenes. These compounds remain in the minority throughout the manuscript. In the multitude of tables, I could find only TWO examples of actual alkenes. Considering this, the title is completely misleading. While the method maybe highly applicable to the synthesis of enol ethers, the authors have not demonstrated It's wide applicability to the corresponding alkenes in any fashion. Therefore I would not find the paper acceptable with this particular title. The authors should really change the title to represent what is shown in the paper if they can't add more alkenes as products.

In terms of the new information, the addition of the Hammett plot is important to demonstrate the effect of the leaving group on the reaction, however they have not performed this Study properly. I Continue to believe that this is important for the paper, to demonstrate the effect of leaving group, but this really needs to be done properly. For a Hammett analysis, the relative rate of reaction of a substituted system versus the corresponding unsubstituted substrate needs to be determined by a competition experiment with the reactions run in the same vessel. Alternatively, the reactions can be run separately and absolute rates compared. As it stands the study does not seem to of been done properly. I would suggest the authors need to do this before acceptance of the paper, and perhaps when done properly, this study will Result in a Hammett plot that gives more information about the reaction.

I would also recommend having the paper reviewed by a native English speaker because there continue to be Issues with grammar and Word choice.

I would be happy to review the paper again, but the Hammett plot needs to be done properly or removed, preferably the former.

Point by point response to the reviewer's comments (NCOMMS-20-07556A)

Reviewer: 1

I appreciate the authors' efforts to address the comments made by all reviewers in the previous round of review. The manuscript appears to be more appropriate for publication on Nature Communications at this point.

Reviewer: 2

I think that the authors have done a really thorough revision of the original manuscript. Our comments are well addressed. I think that the present version should be published, after they fix the following minor typos.

Figure 1c: in the middle, "Y = B(OH₂)" should be "Y = B(OH)₂"

Page 9, Method: line 17: "stirrer bar" should be "stir bar"

We appreciate for the comments of Reviewer 1 and 2, typo errors have been corrected accordingly.

General Procedure for Ni-catalyzed cross-coupling reaction between BzO-DFs (3–5) and arylboronic acids. To a 8 mL vial equipped with a magnetic stir bar was added 2,2-difluorovinyl benzoate **3–5** (1.0 equiv, 0.2 mmol), arylboronic acid **6** (2.0 equiv, 0.4 mmol), K₂CO₃ (3.0 equiv, 0.6 mmol) and deionized water (4.0 equiv, 0.8 mmol) in a nitrogen-filled glovebox.

Reviewer: 3

I have reviewed the revised document and remain unconvinced as to the Authors statement that this work corresponds to a diverse synthesis of gem difluoroalkenes. These compounds remain in the minority throughout the manuscript. In the multitude of tables, I could find only TWO examples of actual alkenes. Considering this, the title is completely misleading. While the method maybe highly applicable to the synthesis of enol ethers, the authors have not demonstrated It's wide applicability to the corresponding alkenes in any fashion. Therefore I would not find the paper acceptable with this particular title. The authors should really change the

title to represent what is shown in the paper if they can't add more alkenes as products.

The comment is well noted, and the title of the manuscript is now changed to "2,2-Difluorovinyl Benzoates as Building Blocks for Diverse Synthesis of *gem*-Difluoroenol Ethers by Ni-Catalyzed Cross Coupling Reaction"

In terms of the new information, the addition of the Hammett plot is important to demonstrate the effect of the leaving group on the reaction, however they have not performed this Study properly. I Continue to believe that this is important for the paper, to demonstrate the effect of leaving group, but this really needs to be done properly. For a Hammett analysis, the relative rate of reaction of a substituted system versus the corresponding unsubstituted substrate needs to be determined by a competition experiment with the reactions run in the same vessel. Alternatively, the reactions can be run separately and absolute rates compared. As it stands the study does not seem to of been done properly. I would suggest the authors need to do this before acceptance of the paper, and perhaps when done properly, this study will Result in a Hammett plot that gives more information about the reaction.

We agree with the Reviewer's comments, the Hammett correlation study has been completely re-done. Please find below for the results:

Competitive experiments – Relative reactivity of the benzoate leaving group

Hammett Correlation Plot

To a 8 mL vial equipped with a magnetic stir, **3a** (0.2 mmol) paired with an equimolar quantity of **3b / 3d / 3f / 3h** (0.2 mmol) was treated with **6c** (0.8 mmol), K₂CO₃ (6.0 equiv, 1.2 mmol) and de-ionized water (8.0 equiv, 1.6 mmol) was mixed in a glovebox. Then a dioxane solution (0.01 M, 4.0 mL) of Ni(COD)₂ (10 mol%, 0.02 mmol) and dppf (12 mol%, 0.024 mmol) was transferred to the vial containing the substrates and reagents inside the glovebox. The reaction mixture was then stirred for 1 hour at 60 °C. To work-up, the reaction mixture was filtered through a pad of Celite[®] and the filtrate was concentrated in *vacuo*. PhCF₃ (0.067 mmol, 8.2 mL, -63.72 ppm) was then added as internal standard for ¹⁹F NMR analysis with CDCl₃ as solvent.

Four sets of competition reaction: **[3a+3b]**, **[3a+3d]**, **[3a+3f]**, **[3a+3h]** were performed in triplicate, and the yields of the remaining substrates were determined by ^{19}F NMR with PhCF_3 (0.067 mmol, 8.2 mL) as an internal standard (-63.72 ppm).

Derivation of the equation used for calculating relative rate information:

For the reaction between n and i competing reactants s_1, s_2, \dots, s_i , the rate equation related to a given species i is:

$$\frac{d s_i}{d t} = k_i s_i [n_{t=0} - (s_{1,t=0} - s_1) - (s_{2,t=0} - s_2) \dots + (s_{i,t=0} - s_i)]$$

Partially integrated form:

$$\ln \left(\frac{s_i}{s_{i,t=0}} \right) = k_i \int [n_{t=0} - (s_{1,t=0} - x_1) - (s_{2,t=0} - s_2) \dots + (s_{i,t=0} - s_i)] dt$$

Solving the equation for two competing species FG and H, the following expression can be obtained for the calculation of relative rate information based on the amount of starting material left:

$$\frac{k_{FG}}{k_H} = \frac{\ln \left(\frac{s_x^{FG}}{s_{x,t=0}^{FG}} \right)}{\ln \left(\frac{s_x^H}{s_{x,t=0}^H} \right)}$$

s_x^{FG} = % yield of **3b / 3d / 3f / 3h** remaining after 1 h of the reaction

$s_{x,t=0}^{FG}$ = % yield of **3b / 3d / 3f / 3h** initially = 100%

s_x^H = % yield of **3a** remaining after 1 h of the reaction

$s_{x,t=0}^H$ = % yield of **3a** initially = 100%

To perform the free-energy study, the relative rate values were fitted into the following Hammett equation:

$$\log \left(\frac{k_{FG}}{k_H} \right) = \rho \sigma$$

ρ = reaction constant, σ = Hammett substituent constant (*para*)

- ¹⁹F NMR spectra of pure **3a** and pure **3b**

- ¹⁹F NMR spectra of (reaction mixture "3a + 3b" – trial 1)

- ¹⁹F NMR spectra of (reaction mixture “3a + 3b” – trial 2)

- ¹⁹F NMR spectra of (reaction mixture “3a + 3b” – trial 3)

- ¹⁹F NMR spectra of pure **3a** and pure **3d**

- ¹⁹F NMR spectra of (reaction mixture "3a + 3d" – trial 1)

- ¹⁹F NMR spectra of (reaction mixture “3a + 3d” – trial 2)

- ¹⁹F NMR spectra of (reaction mixture “3a + 3d” – trial 3)

- ¹⁹F NMR spectra of pure **3a** and pure **3f**

- ¹⁹F NMR spectra of (reaction mixture "3a + 3f" – trial 1)

- ¹⁹F NMR spectra of (reaction mixture "3a + 3f" – trial 2)

- ¹⁹F NMR spectra of (reaction mixture "3a + 3f" – trial 3)

- ¹⁹F NMR spectra of pure **3a** and pure **3h**

- ¹⁹F NMR spectra of (reaction mixture "3a + 3h" – trial 1)

- ¹⁹F NMR spectra of (reaction mixture "3a + 3h" – trial 2)

- ¹⁹F NMR spectra of (reaction mixture "3a + 3h" – trial 3)

Result summary for [3a (H) + 3b (Me)]

trial	3a remaining	3b remaining	$\frac{k_{Me}}{k_H}$	$\log\left(\frac{k_{Me}}{k_H}\right)$	Avg. $\log\left(\frac{k_{Me}}{k_H}\right)$
1	51%	81%	0.31	-0.50	
2	46%	77%	0.34	-0.47	-0.45
3	45%	72%	0.41	-0.39	

Result summary for [3a (H) + 3d (Cl)]

trial	3a remaining	3d remaining	$\frac{k_{Cl}}{k_H}$	$\log\left(\frac{k_{Cl}}{k_H}\right)$	Avg. $\log\left(\frac{k_{Cl}}{k_H}\right)$
1	40%	60%	0.59	-0.25	
2	50%	65%	0.62	-0.21	-0.18
3	48%	54%	0.84	-0.08	

Result summary for [3a (H) + 3f (F)]

trial	3a remaining	3f remaining	$\frac{k_F}{k_H}$	$\log\left(\frac{k_F}{k_H}\right)$	Avg. $\log\left(\frac{k_F}{k_H}\right)$
1	39%	48%	0.78	-0.11	
2	50%	59%	0.76	-0.12	-0.11
3	38%	46%	0.80	-0.10	

Result summary for [3a (H) + 3h (tBu)]

trial	3a remaining	3h remaining	$\frac{k_{tBu}}{k_H}$	$\log\left(\frac{k_{tBu}}{k_H}\right)$	Avg. $\log\left(\frac{k_{tBu}}{k_H}\right)$
1	64%	86%	0.34	-0.47	
2	67%	88%	0.32	-0.49	-0.51
3	71%	91%	0.28	-0.56	

Hammett Correlation Plot

FG	pKa of FG-C ₆ H ₄ CO ₂ H	σ	Avg. $\log\left(\frac{k_{FG}}{k_H}\right)$
^t Bu	4.57	-0.20	-0.51
Me	4.38	-0.17	-0.45
H	4.20	0	0
F	4.14	0.06	-0.11
Cl	3.97	0.23	-0.17

Apparently, the free-energy plot reveals a concave relationship between $\log(k_{FG}/k_H)$ and the Hammett sigma constants. The origin of this non-linear relationship remains unclear. We speculate that two opposing interactions maybe in operation at the transition states. Yet, a separate study is warranted to delineate clearly this finding.

REVIEWER COMMENTS

Reviewer #3 (Remarks to the Author):

The change to the title is a good one and is now more representative of the reaction the authors are describing.

In terms of the Hammett study, I know that the authors normally use a 2:1 ratio of boronic acid to difluoroenol ester, but the way a Hammett study is normally run, one would use an excess of the reactant whose electronic effects are being checked. However, I see that the authors have run the reaction to low conversion, by stopping it after one hour, which gives about 50% conversion, so this is a good alternative.

However, it is not standard practice to show a curved line for Hammett studies. Instead, the authors really need to show two broken lines, so more of a V-shape than a curved line. When I look at their data in this way, it appears that they have a V-shaped plot with a slightly steeper slope for substrates more electron donating than hydrogen. Once they break this down into two sections, they can derive an R squared factor for each line. However, I think part of the problem is that they have a very low spread of sigma values. So tBu as the most electron donating and Cl as the most withdrawing. This is a very narrow range, and makes interpretation a problem, but at the minimum they need to actually divide this up into two sections, use straight lines, and put numbers on their axes for the x and y axes.

In addition, I know that this is more complicated to interpret, but concave up versus concave down plots have known interpretations, either change in mechanism or more change in rate determining step, so with a small look at the literature this could be clarified, but considering that this is a nature communication, I would encourage them to introduce one more electron donating substrate and one more electron withdrawing.

Finally, for their plot of yield versus pKa, they also need to use actual straight lines and introduce numbers on their x and y axes.

Once these issues have been corrected, I would find the paper acceptable for publication.

Point by point response to the reviewer's comments (NCOMMS-20-07556A)

Reviewer: 3

In terms of the Hammett study, I know that the authors normally use a 2:1 ratio of boric acid to difluoroenol ester, but the way a Hammett study is normally run, one would use an excess of the reactant whose electronic effects are being checked. However, I see that the authors have run the reaction to low conversion, by stopping it after one hour, which gives about 50% conversion, so this is a good alternative.

However, it is not standard practice to show a curved line for Hammett studies. Instead, the authors really need to show two broken lines, so more of a V-shape than a curved line. When I look at their data in this way, it appears that they have a V-shaped plot with a slightly steeper slope for substrates more electron donating than hydrogen. Once they break this down into two sections, they can derive an R squared factor for each line. However, I think part of the problem is that they have a very low spread of sigma values. So tBu as the most electron donating and Cl as the most withdrawing. This is a very narrow range, and makes interpretation a problem, but at the minimum they need to actually divide this up into two sections, use straight lines, and put numbers on their axes for the x and y axes.

In addition, I know that this is more complicated to interpret, but concave up versus concave down plots have known interpretations, either change in mechanism or change in rate determining step, so with a small look at the literature this could be clarified, but considering that this is a nature communication, I would encourage them to introduce one more electron donating substrate and one more electron withdrawing.

Finally for their plot of yield versus pKa, they also need to use actual straight lines and introduce numbers on their x and y axes.

We revisited the Hammett study and added two more substrates. A linear plot was obtained when a wider spread of sigma values was considered. Attached please find the updated result.

To a 8 mL vial equipped with a magnetic stir, **3a** (0.2 mmol) paired with an equimolar quantity of **3b / 3c / 3d / 3f / 3h / 3i** (0.2 mmol) was treated with **6c** (0.8 mmol), K_2CO_3 (6.0 equiv, 1.2 mmol) and de-ionized water (8.0 equiv, 1.6 mmol) was mixed in a glovebox. Then a dioxane solution (0.01 M, 4.0 mL) of $\text{Ni}(\text{COD})_2$ (10 mol%, 0.02 mmol) and dppf (12 mol%, 0.024 mmol) was transferred to the vial containing the substrates and reagents inside the glovebox. The reaction mixture was then stirred for 1 hour at $60\text{ }^\circ\text{C}$. To work-up, the reaction mixture was filtered through a pad of Celite[®] and the filtrate was concentrated in *vacuo*. PhCF_3 (0.067 mmol, $8.2\text{ }\mu\text{L}$, -63.72 ppm) was then added as internal standard for ^{19}F NMR analysis with CDCl_3 as solvent.

Four sets of competition reaction: [**3a+3b**], [**3a+3d**], [**3a+3f**], [**3a+3h**] were performed in triplicate, and the yields of the remaining substrates were determined by ¹⁹F NMR with PhCF₃ (0.067 mmol, 8.2 μL) as an internal standard (-63.72 ppm).

Derivation of the equation used for calculating relative rate information:

For the reaction between *n* and *i* competing reactants *s*₁, *s*₂, ...*s*_{*i*}, the rate equation related to a given species *i* is:

$$\frac{d_{s_i}}{d_t} = k_i s_i [n_{t=0} - (s_{1,t=0} - s_1) - (s_{2,t=0} - s_2) \dots + (s_{i,t=0} - s_i)]$$

Partially integrated form:

$$\ln \left(\frac{s_i}{s_{i,t=0}} \right) = k_i \int [n_{t=0} - (s_{1,t=0} - x_1) - (s_{2,t=0} - s_2) \dots + (s_{i,t=0} - s_i)] dt$$

Solving the equation for two competing species FG and H, the following expression can be obtained for the calculation of relative rate information based on the amount of starting material left:

$$\frac{k_{FG}}{k_H} = \frac{\ln \left(\frac{s_x^{FG}}{s_{x,t=0}^{FG}} \right)}{\ln \left(\frac{s_x^H}{s_{x,t=0}^H} \right)}$$

s_x^{FG} = % yield of **3b** / **3d** / **3f** / **3h** remaining after 1 h of the reaction

$s_{x,t=0}^{FG}$ = % yield of **3b** / **3d** / **3f** / **3h** initially = 100%

s_x^H = % yield of **3a** remaining after 1 h of the reaction

$s_{x,t=0}^H$ = % yield of **3a** initially = 100%

To perform the free-energy study, the relative rate values were fitted into the following Hammett equation:

$$\log \left(\frac{k_{FG}}{k_H} \right) = \rho \sigma$$

ρ = reaction constant, σ = Hammett substituent constant (*para*)

- ¹⁹F NMR spectra of pure **3a** and pure **3b**

- ¹⁹F NMR spectra of (reaction mixture "3a + 3b" – trial 1)

- ¹⁹F NMR spectra of (reaction mixture “3a + 3b” – trial 2)

- ¹⁹F NMR spectra of (reaction mixture “3a + 3b” – trial 3)

- ^{19}F NMR spectra of pure **3a** and pure **3c**

- ^{19}F NMR spectra of (reaction mixture "3a + 3c" – trial 1)

- ¹⁹F NMR spectra of (reaction mixture “3a + 3c” – trial 2)

- ¹⁹F NMR spectra of (reaction mixture “3a + 3c” – trial 3)

- ^{19}F NMR spectra of pure **3a** and pure **3d**

- ^{19}F NMR spectra of (reaction mixture "3a + 3d" – trial 1)

- ¹⁹F NMR spectra of (reaction mixture “3a + 3d” – trial 2)

- ¹⁹F NMR spectra of (reaction mixture “3a + 3d” – trial 3)

- ^{19}F NMR spectra of pure **3a** and pure **3f**

- ^{19}F NMR spectra of (reaction mixture "3a + 3f" – trial 1)

- ¹⁹F NMR spectra of (reaction mixture “3a + 3f” – trial 2)

- ¹⁹F NMR spectra of (reaction mixture “3a + 3f” – trial 3)

- ^{19}F NMR spectra of pure **3a** and pure **3h**

- ^{19}F NMR spectra of (reaction mixture “**3a** + **3h**” – trial 1)

- ¹⁹F NMR spectra of (reaction mixture “3a + 3h” – trial 2)

- ¹⁹F NMR spectra of (reaction mixture “3a + 3h” – trial 3)

- ¹⁹F NMR spectra of pure **3a** and pure **3i**

- ¹⁹F NMR spectra of (reaction mixture “**3a** + **3i**” – trial 1)

- ¹⁹F NMR spectra of (reaction mixture “3a + 3i” – trial 2)

- ¹⁹F NMR spectra of (reaction mixture “3a + 3i” – trial 3)

Result summary for [3a (H) + 3b (Me)]

trial	3a remaining	3b remaining	$\frac{k_{\text{Me}}}{k_{\text{H}}}$	$\log\left(\frac{k_{\text{Me}}}{k_{\text{H}}}\right)$	Avg. $\log\left(\frac{k_{\text{Me}}}{k_{\text{H}}}\right)$
1	51%	81%	0.31	-0.50	
2	46%	77%	0.34	-0.47	-0.45
3	45%	72%	0.41	-0.39	

Result summary for [3a (H) + 3c (OMe)]

trial	3a remaining	3d remaining	$\frac{k_{\text{OMe}}}{k_{\text{H}}}$	$\log\left(\frac{k_{\text{OMe}}}{k_{\text{H}}}\right)$	Avg. $\log\left(\frac{k_{\text{OMe}}}{k_{\text{H}}}\right)$
1	58%	91%	0.17	-0.76	
2	54%	82%	0.32	-0.49	-0.68
3	57%	89%	0.21	-0.68	

Result summary for [3a (H) + 3d (Cl)]

trial	3a remaining	3d remaining	$\frac{k_{\text{Cl}}}{k_{\text{H}}}$	$\log\left(\frac{k_{\text{Cl}}}{k_{\text{H}}}\right)$	Avg. $\log\left(\frac{k_{\text{Cl}}}{k_{\text{H}}}\right)$
1	40%	60%	0.59	-0.25	
2	50%	65%	0.62	-0.21	-0.18
3	48%	54%	0.84	-0.08	

Result summary for [3a (H) + 3f (F)]

trial	3a remaining	3f remaining	$\frac{k_{\text{F}}}{k_{\text{H}}}$	$\log\left(\frac{k_{\text{F}}}{k_{\text{H}}}\right)$	Avg. $\log\left(\frac{k_{\text{F}}}{k_{\text{H}}}\right)$
1	39%	48%	0.78	-0.11	
2	50%	59%	0.76	-0.12	-0.11
3	38%	46%	0.80	-0.10	

Result summary for [3a (H) + 3h (tBu)]

trial	3a remaining	3h remaining	$\frac{k_{tBu}}{k_H}$	$\log\left(\frac{k_{tBu}}{k_H}\right)$	Avg. $\log\left(\frac{k_{tBu}}{k_H}\right)$
1	64%	86%	0.34	-0.47	
2	67%	88%	0.32	-0.49	-0.51
3	71%	91%	0.28	-0.56	

Result summary for [3a (H) + 3i (CF₃)]

trial	3a remaining	3i remaining	$\frac{k_{CF_3}}{k_H}$	$\log\left(\frac{k_{CF_3}}{k_H}\right)$	Avg. $\log\left(\frac{k_{CF_3}}{k_H}\right)$
1	64%	55%	1.34	0.13	
2	59%	56%	1.10	0.41	0.27
3	61%	59%	1.07	0.28	

Hammett Correlation Plot

FG	pKa of FG-C ₆ H ₄ CO ₂ H	σ	Avg. $\log\left(\frac{k_{FG}}{k_H}\right)$
OMe	4.57	-0.27	-0.68
tBu	4.57	-0.20	-0.51
Me	4.38	-0.17	-0.45
H	4.20	0	0
F	4.14	0.06	-0.11
Cl	3.97	0.23	-0.17
CF ₃	3.69	0.54	0.27

Next, we investigated the effects of pK_a of the leaving groups on the construction of BzO-DFs and the corresponding coupling with boronic acids. The effects of pK_a range for the synthesis of **3** and **7c** was found to be 3.97–4.57.

- ¹⁹F NMR spectra of pure **3a** and pure **3b**

- ¹⁹F NMR spectra of (reaction mixture "3a + 3b" – trial 1)

- ¹⁹F NMR spectra of (reaction mixture "3a + 3b" – trial 2)

- ¹⁹F NMR spectra of (reaction mixture "3a + 3b" – trial 3)

- ¹⁹F NMR spectra of pure **3a** and pure **3d**

- ¹⁹F NMR spectra of (reaction mixture "3a + 3d" – trial 1)

- ¹⁹F NMR spectra of (reaction mixture "3a + 3d" – trial 2)

- ¹⁹F NMR spectra of (reaction mixture "3a + 3d" – trial 3)

- ^{19}F NMR spectra of pure **3a** and pure **3f**

- ^{19}F NMR spectra of (reaction mixture "3a + 3f" – trial 1)

- ¹⁹F NMR spectra of (reaction mixture "3a + 3f" – trial 2)

- ¹⁹F NMR spectra of (reaction mixture "3a + 3f" – trial 3)

- ¹⁹F NMR spectra of pure **3a** and pure **3h**

- ¹⁹F NMR spectra of (reaction mixture "3a + 3h" – trial 1)

- ¹⁹F NMR spectra of (reaction mixture "3a + 3h" – trial 2)

- ¹⁹F NMR spectra of (reaction mixture "3a + 3h" – trial 3)

Result summary for [3a (H) + 3b (Me)]

trial	3a remaining	3b remaining	$\frac{k_{Me}}{k_H}$	$\log\left(\frac{k_{Me}}{k_H}\right)$	Avg. $\log\left(\frac{k_{Me}}{k_H}\right)$
1	51%	81%	0.31	-0.50	
2	46%	77%	0.34	-0.47	-0.45
3	45%	72%	0.41	-0.39	

Result summary for [3a (H) + 3d (Cl)]

trial	3a remaining	3d remaining	$\frac{k_{Cl}}{k_H}$	$\log\left(\frac{k_{Cl}}{k_H}\right)$	Avg. $\log\left(\frac{k_{Cl}}{k_H}\right)$
1	40%	60%	0.59	-0.25	
2	50%	65%	0.62	-0.21	-0.18
3	48%	54%	0.84	-0.08	

Result summary for [3a (H) + 3f (F)]

trial	3a remaining	3f remaining	$\frac{k_F}{k_H}$	$\log\left(\frac{k_F}{k_H}\right)$	Avg. $\log\left(\frac{k_F}{k_H}\right)$
1	39%	48%	0.78	-0.11	
2	50%	59%	0.76	-0.12	-0.11
3	38%	46%	0.80	-0.10	

Result summary for [3a (H) + 3h (tBu)]

trial	3a remaining	3h remaining	$\frac{k_{tBu}}{k_H}$	$\log\left(\frac{k_{tBu}}{k_H}\right)$	Avg. $\log\left(\frac{k_{tBu}}{k_H}\right)$
1	64%	86%	0.34	-0.47	
2	67%	88%	0.32	-0.49	-0.51
3	71%	91%	0.28	-0.56	

Hammett Correlation Plot

FG	pKa of FG-C ₆ H ₄ CO ₂ H	σ	Avg. $\log\left(\frac{k_{FG}}{k_H}\right)$
^t Bu	4.57	-0.20	-0.51
Me	4.38	-0.17	-0.45
H	4.20	0	0
F	4.14	0.06	-0.11
Cl	3.97	0.23	-0.17

Apparently, the free-energy plot reveals a concave relationship between $\log(k_{FG}/k_H)$ and the Hammett sigma constants. The origin of this non-linear relationship remains unclear. We speculate that two opposing interactions maybe in operation at the transition states. Yet, a separate study is warranted to delineate clearly this finding.